# Sustainable Tourism in Protected Areas—The Case of the Vršac Mountains Outstanding Natural Landscape, Vojvodina Province (Northern Serbia)

Igor Trišić [1,*], Florin Nechita [2], Vladica Ristić [3], Snežana Štetić [4,5], Marija Maksin [6] and Ioana Anisa Atudorei [2]

1    Faculty of Geography, University of Belgrade, Studentski Trg 3/III, 11000 Belgrade, Serbia
2    Department of Social and Communication Sciences, Transilvania University of Brasov, 29, Eroilor Bd., 500036 Brasov, Romania; florin.nechita@unitbv.ro (F.N.); ioana.atudorei@unitbv.ro (I.A.A.)
3    Faculty of Applied Ecology "Futura", Metropolitan University, Požeška 83, 11030 Belgrade, Serbia; vladicar011@gmail.com
4    The College of Tourism Belgrade, Bulevar Zorana Đinđića 152a, 11070 Belgrade, Serbia; snezana.stetic@gmail.com
5    Balkan Network of Tourism Experts, 11000 Belgrade, Serbia
6    Institute of Architecture and Urban & Spatial Planning of Serbia, Bulevar Kralja Aleksandra 73/II, 11000 Belgrade, Serbia; micic70a@yahoo.com
*    Correspondence: trisici@hotmail.com; Tel.: +381-641-431-375

**Abstract:** The sustainable development of tourism in protected areas and the planning of its development is extremely important for mountain regions. The local population and tourists have a very important role in this process. Therefore, surveys of the local population and visitor satisfaction represent the basis of this research. The Vršac Mountains Outstanding Natural Landscape (ONL) could represent a significant destination for sustainable tourism. This mountainous area is characterized by a favorable geographical position, a diversity of natural and social factors, rare flora and fauna, and the rich ethno-social heritage of the local population. It is in a very favorable geographical position, and the proximity of the big cities Serbia and Romania, as well as many other factors, are important for tourism development in this area. A quantitative methodology was used for the purposes of this paper during our research. The purpose of this research was to investigate the influence of tourism development on the satisfaction of residents and visitors of the ONL by applying a survey technique, with the help of a questionnaire as a research instrument. A total of 1419 respondents were surveyed (789 residents and 630 visitors). The research results show that the ecological and socio-cultural dimensions of sustainability have the greatest importance for the respondents, and that these two dimensions of sustainability have the greatest impact on sustainable tourism in this protected area. Our main research hypothesis, which states that sustainable tourism has a positive impact on the satisfaction of residents and visitors, is fully confirmed. These data could be significant for tourism planning and the management of protected areas.

**Keywords:** prism of sustainability; tourism development; planning; nature-based tourism; events

## 1. Introduction

This study represents a continuation of extensive research on sustainable tourism within a spatial entity such as the Province of Vojvodina. By studying each protected area separately in this part of Serbia, the authors aim to create a complete picture of the level of nature protection in Vojvodina. By surveying visitors and local residents regarding their views on the development of sustainable tourism, a complete picture of the impact of environmental protection and tourism development, their economic benefits, and the role of local institutions is obtained. The subject of research in this paper is protected areas in which there are certain tourist activities whose goal is nature preservation. In addition,

this study is significant because its results could serve in the planning and development of specific sustainable forms of tourism. These forms of tourism can directly and indirectly achieve ecological, economic, socio-cultural, and social benefits for the local community in the development of sustainable tourism. The protected area of the Vršac Mountains Outstanding Natural Landscape (ONL) is in the Autonomous Province of Vojvodina (in Northern part of Serbia), on the territory of eight urban settlements: Sočica, Jablanka, Mesić, Vršac, Veliko Središte, Malo Središte, Gudurica, and Markovac. A good connection with urban outbound (emission) centers is a special advantage of its geographical position and could be beneficial for the development of tourism. The geographical position of the ONL has influenced the development of flora and fauna. This region is characterized by diverse flora (over 1000 species), some of which are endemic. In addition, this area is characterized by a variety of fauna. In addition to its many significant natural features, 14 archaeological sites have been discovered in the ONL area. Some of them are also recorded on the archaeological maps of Europe, which will be the subject of our future research. The above-mentioned natural, anthropogenic, and social factors represent significant potential for tourism development [1].

In this protected area, there are different and very specific ecosystems inhabited by characteristic plant and animal species. Therefore, the planning of tourism development in this area should be informed by research on the complete natural resources, with emphasis on the best opportunities for the protection of the area and the existing plant and animal species. The natural factors that are present in the ONL enable the development of specific forms of tourism, which should represent the basis for the development of its own tourism. The research conducted in this paper will provide some answers to questions about the possibility of valorizing this protected area.

To date, the following forms of tourism have been developed: nature-based tourism, ecotourism, scientific-research tourism, trips, a school in nature, sports/recreational tourism, etc. In the area of the ONL, as well as in other areas of Vojvodina, there are regions where vines are grown. This area is called "Vršac Vineyards". It covers an area of 1700 ha. Grape varieties are grown in Župljanka, Riesling Italico, Gutedel weisser—Chasselas, and Kreaca, from which top wines are produced. This has influenced the development of numerous wineries in the ONL area, which are valorized through tourism and which are visited by domestic and foreign tourists. In recent years, this region has received an increasing number of tourists, and its characteristics allow the ONL area to be included in the wine routes of Serbia [2].

The diversity of the population of Vojvodina and its multiculturalism influence the creation of special ethno-social characteristics in each individual area. The ethno-social values of the population that inhabits the ONL area are reflected through economic and social customs, native music, gastronomy, domestic crafts, and costumes. Together they represent important tourism potential that can be included in tourist offerings [1,3].

By combining natural and social factors in the promotion and creation of a tourism product, this protected area could be positioned as an important destination for sustainable tourism. The active involvement of local residents and visitors in tourism development would create a basis of human resources that could extremely positively influence the formation of conditions for the ONL to become a developed tourist destination. This is very important from the aspect of tourism valorization [4] of the development of sustainable tourism [5,6]. Achieving ecological, economic, socio-cultural, and institutional benefits from proper tourism development represents the concept of sustainable tourism, which is also the objective of the research in this paper [7–9].

This paper investigates the satisfaction of ONL residents and visitors, which is affected by sustainable tourism, through four dimensions of sustainability. These are the ecological, socio-cultural, economic, and institutional dimensions of sustainability.

By analyzing the level of satisfaction, we can conceptualize the state of sustainable tourism in this protected area and the impacts that the ONL has on tourism, in addition to the satisfaction of residents and visitors [10]. The aim of this study is to examine the function of sustainable tourism within the protected area with the help of a quantitative methodology. This means that the focus of the study is on measuring the importance of individual dimensions of sustainability in the overall development of tourism. The initial basis for the research is defined in the hypotheses. The main hypothesis (H1) in the paper is that sustainable tourism has a significant impact on the satisfaction of visitors and residents. The auxiliary hypotheses (H1.1, H1.2 . . . H1.4) concern the examination of the individual impacts of the four dimensions of sustainability (ecological, economic, socio-cultural, and institutional) on the sustainable development of tourism within the protected area. This information can provide important answers regarding the state of sustainable tourism in this protected area [11,12].

Using a random sampling method, a total of 1419 respondents were surveyed regarding sustainable tourism in the ONL (789 residents and 630 visitors). The published data were analyzed using the statistical program SPSS v.21 Software (IBM, Armonk, NY, USA).

The main conclusions of this research emphasize that natural and social factors within the protected area are extremely important when planning the development of tourism. In addition to the ecological dimension, the socio-cultural dimension of sustainability also plays a very important role. This is reflected in emphasizing the importance of the local population in the development of tourism within the protected area. At the end of this paper, important concluding considerations emphasize the positive impact of sustainable tourism development on residents and visitors due to its influence on their experiences.

The authors' future research will include an examination of the level of sustainable tourism development in certain protected areas of Romania, after which a comparative analysis will be applied to the results of this research. Therefore, it is important to perform a comparative analysis of factors that may be significant for tourism development. The results obtained in this way can be used to develop tourism development strategies for protected border areas or protected areas that are important for certain regions in the world [13]. In addition, the results will aim to strengthen border cooperation between the Republic of Romania and the Republic of Serbia, in terms of tourism development and in numerous other fields.

In this research, the significance of sustainable tourism for the development of a destination or region is emphasized. This can be seen through the Prism of Sustainability, which means that the sustainable development of tourism should aim to achieve positive and long-term results. The originality of this research lies in the analysis of the current level of development of sustainable tourism, whereby the perceived attitudes of residents and visitors, as well as their degree of satisfaction with sustainable tourism, were assessed. Another original aspect of this research is reflected in the selection of the subject of research, which is a protected mountain area that has many different natural and social factors, and potential for the development of tourism. Our research results can be used to determine the strategy of tourism development and the management of this protected area. In addition, these results can serve in planning the development of tourism in other protected areas in the country or in a wider area. The main implication of our results is the importance of the ecological awareness of visitors and the population, whose primary wishes are for these areas to be protected. Tourism is then highlighted as a supporting activity.

The main chapters in this paper are as follows: the Abstract, which provides a brief description of the research and the results; the Introduction with a description of the research objective, methods used, research focus, implications, limitations, and expected scientific contributions; a Literature Review describing research that represents the fundamental basis of this study and that serves in defining our research model and research area; the Methodology, which provides a description of the methodology; the Results, which clearly highlight the obtained values and the research results; the Discussion, in which the ob-

tained results are discussed and their practical scientific applications are examined; the Conclusions; and Limitations and Future Research.

During the period of data collection, there was a limitation (namely the COVID-19 pandemic) that, at different intervals, led to an increased number of patients, which affected the realization of personal contact with respondents, and to a greater extent, the number of visitors to the protected area.

## 2. Literature Review

By analyzing the existing available literature, which focuses on the study of sustainable tourism within different protected areas, it can be concluded that there are certain shortcomings related to data collection during field research. In a large number of papers that apply quantitative methodology, the data are collected only from one group of respondents, i.e., residents. Although their role is significant in sustainable development, residents' answers can also have a subjective connotation regarding their perceptions. This may affect the objectivity of the study, which aims to evaluate and measure the value of certain phenomena in the planning and development of tourism. Therefore, in this research, the authors collected data from two groups of respondents. In addition to residents, the perceived views of visitors on the level of development and importance of sustainable tourism were also examined. With the research performed in this way, the authors obtained original and credible results that could be used and compared.

Research on sustainable tourism in protected areas can provide significant insight into the importance of implementing protected areas into tourism [14]. The focus of numerous studies is on the examination of the importance of socio-economic development in connection with the ecological factors of protected areas and the preservation of biodiversity [15–17]. The specificity of sustainable tourism studies implies the inclusion of all interested parties in planning and development, and takes into account all the specificities of geographical features and different conditions of the local environment [18]. The importance of studying protected area tourism lies in finding the best scenario for sustainable tourism development [19]. At the same time, this postulate led us to examine sustainable tourism development in this paper.

The literature in the field of sustainable tourism development in protected areas defines tourism development as a complex system in which numerous goals must be achieved [20] in order to reach the ultimate goal—the development of sustainable tourism [21]. Environmental, socio-cultural, economic, and institutional objectives stand out as the most important [22,23].

The development of tourism in protected areas aims to achieve sustainability through various plans, measures, and activities [24]. The improvement of ecological principles [25,26] and the satisfaction of tourists and local communities are the main aims, and can affect the realized income and the direction of tourism development in protected areas, and the preservation of nature within them [27,28].

Tourists are no longer satisfied with passive observation of the environment, and they insist on active involvement in their trips. In doing so, the emphasis is placed on the sustainability of development, the authenticity of the content, and the entire experience when traveling and staying in protected areas. This requires getting to know the environment, population, culture, attractiveness, and an interactive relationship with these elements. Accordingly, the inclusion of scientific research, wine, educational, excursions, sports/recreation, and rural tourism, along with ecotourism, would significantly affect the promotion of protected areas as tourist destinations and strengthen overall tourist offerings [26,29].

The development of various forms of tourism in protected areas can contribute to social, cultural, economic, and ecological benefits through various influences. Therefore, the proper management of all attractions within the protected area is very important. Among the significant activities in the development of tourism in these areas, the promotion of the protection of natural and anthropogenic resources, the respect of previous protection,

the management of existing resources, the realization of economic benefits, etc., stand out [30,31]. This represents a significant postulate of sustainability in which the ecological aspect stands out as an initiator of other sustainable dimensions, such as economic, sociocultural, and institutional [2].

In addition to ecotourism, special forms of tourism, such as adventure, educational, sports/recreational, and event tourism, are mentioned in the literature as significant drivers of sustainability within protected areas. These forms can contribute to the strengthening of the natural and anthropogenic factors of sustainable tourism development within protected areas [32]. In addition to enhancing the quality of the destination, these forms of tourism aim to bring economic and socio-cultural benefits. A destination with degraded ecological values is not attractive to visitors [33–36].

In addition to the basic attractive tourism factors, sustainable tourism development in protected areas can also depend on the protection of the space, the intensity of space use, and the carrying capacity of the space [37], tourism development, sociocultural influences, the contribution of tourism to the local economy, development control, waste management, etc. [25,27,33,38–40]. The importance of various factors in the development of a tourist destination can be more precisely determined, measured, and monitored using indicators of sustainable tourism [41–43]. Furthermore, apart from environmental preservation, the level of space degradation caused by the increased number of visitors has a considerable impact. Therefore, the correlation between the number of tourists [44] and the number of inhabitants in the protected area and the protective zone around the studied area must be taken into account. The study of air pollution and hydro-pollution, harmful anthropogenic activities, the costs of the reconstruction and protection of damaged areas, social impacts, and economic efficiency, which can be managed through the development of controlled forms of tourism, are only some additional factors that must be given special attention in order to create tourism products in protected areas [33,45].

Models applied when researching the impacts of specific tourism destinations with natural and cultural heritage on sustainable tourism development were identified as the most important factors of tourism development within protected areas, as well as those that are related to the natural and social elements of the destination and its built-up environment. Moreover, other extremely important factors are the influence of space and the living world on tourism development, the role of the local community in sustainable development [46], the implementation of legal measures, the possibility of developing different forms of tourism, the improvement of local traditions and culture, the carrying capacity of the space, the exploitation of resources, etc. [25,27,39,44,47–52].

The objective of the research of Huayhuaca et al. [53] was to examine the importance of sustainable tourism to the local population in the Frankenwald Nature Park in Germany. Their research was carried out using the Prism of Sustainability Model, which was designed to measure the respondents' perception of four dimensions of sustainability: environmental, economic, socio-cultural, and institutional. The scientific contribution of this research is to provide significant information about sustainable tourism development in protected area, which can be used for examining sustainable tourism in other protected areas in the world.

A study by Cottrell et al. [54] was based on examining the impact of sustainable development on visitors in two protected areas. The research methodology was also conceived according to the PoS model. The respondents rated these two sustainability factors as the most important factors for the development of tourism in the two examined protected areas, which represents the most significant results of the research. The model used in this research served the authors in other case studies examining the Prism of Sustainability.

### 3. Research Area

In order to determine the function of protected areas in sustainable tourism development over the most significant scope and the widest span, the authors applied a research model to the selected research area. By including this protected area in the focus of this sustainable tourism study, the range of the authors' protected research area in Vojvodina, as one territorially unique spatial unit in the context of sustainable tourism development, has expanded.

The ONL area is located in the southeastern part of the Autonomous Province of Vojvodina (Figure 1) and is part of the Vršac Mountains, with the highest peak (the Gudurica Peak (641 m) [1]. This protected area is located on the territory of eight inhabited settlements: Sočica, Jablanka, Mesić, Vršac, Veliko Središte, Malo Središte, Gudurica, and Markovac. The ONL covers an area of 5328.86 ha (a total of 31.35% of the area of the Vršac Mountains). Within the protected area, the first, second, and third degrees of protection regimes have been established. According to the IUCN, the ONL belongs to the fourth category—Habitat and Species Management Area [55,56].

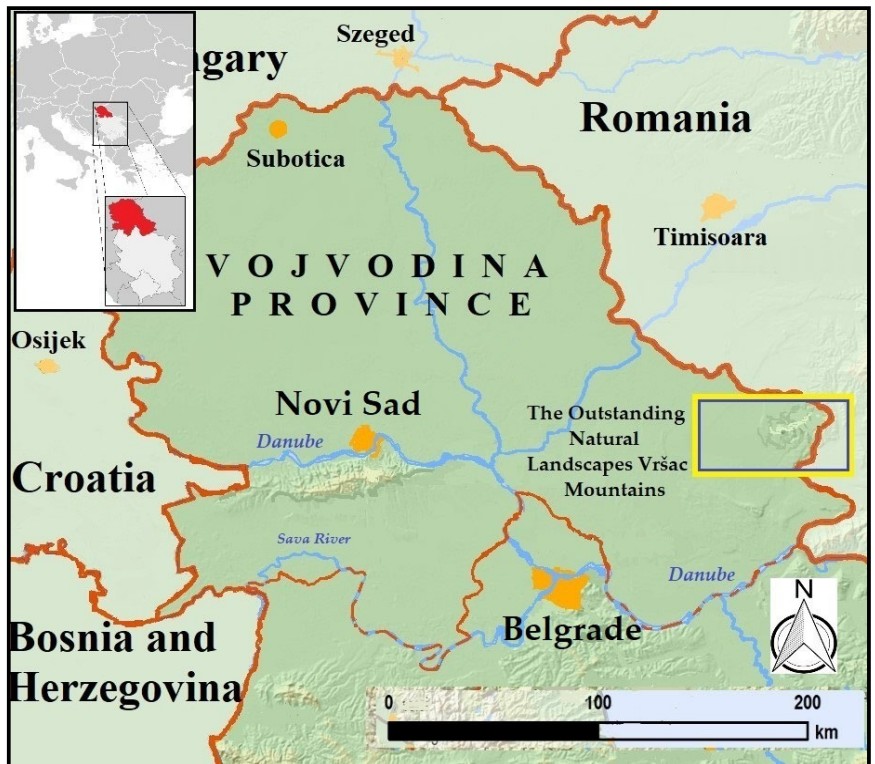

**Figure 1.** Study area. Source: created by the author.

In the area of the Vršac Mountains, 130 species of birds have been registered. The following are the most important ornithological representatives: the lesser spotted eagle (*Aquila pomarina*), the European honey buzzard (*Pernis opivarius*), the Levant sparrowhawk (*Accipiter brevipes*), the short-toed snake eagle (*Circaetus gallicius*), the white-backed woodpecker (*Dendrocopos leucotos*), and others. Since 1989, the Vršac Mountains (with an ornithological zone area of 10,500 ha) have been included in the world register of internationally important bird habitats (IBA area). The nesting of the ferruginous duck (*Aythya nyroca*) particularly stands out in this area. This area, which is rich in birds, is also home to different species of insects, mammals, reptiles, and important endemic representatives of flora [57,58].

## 4. Methodology

In the examination of the function of the ONL as a protected area in sustainable tourism development, *the Prism of Sustainability* model was used. This version was designed according to previously used models in the study of sustainable tourism in some other protected areas [53,54,59]. This research model was used in the constitution of the research technique and instrument. In addition, the research model was used in the selection of statistical methods for data processing and analysis. Residents and visitors of the protected area were surveyed with the help of a questionnaire. The questionnaire was designed to contain 17 items, grouped into four dimensions of sustainability, according to the downloaded questionnaire models. Respondents anonymously expressed their perceived views on the statements made in the questionnaire. In addition to the above, the questionnaire also contained 4 items concerning the influence of sustainable tourism on the pleasure of respondents. Regression analysis was used to measure satisfaction, also based on the accepted research model (PoS) [53,54]. This research model was adapted to studies of the attitudes of respondents in this protected area. By applying the PoS Model in this research, we were able to measure the respondents' perceived attitudes regarding four dimensions of sustainability: ecological, economic, socio-cultural, and institutional sustainability [60–62] (Figure 2).

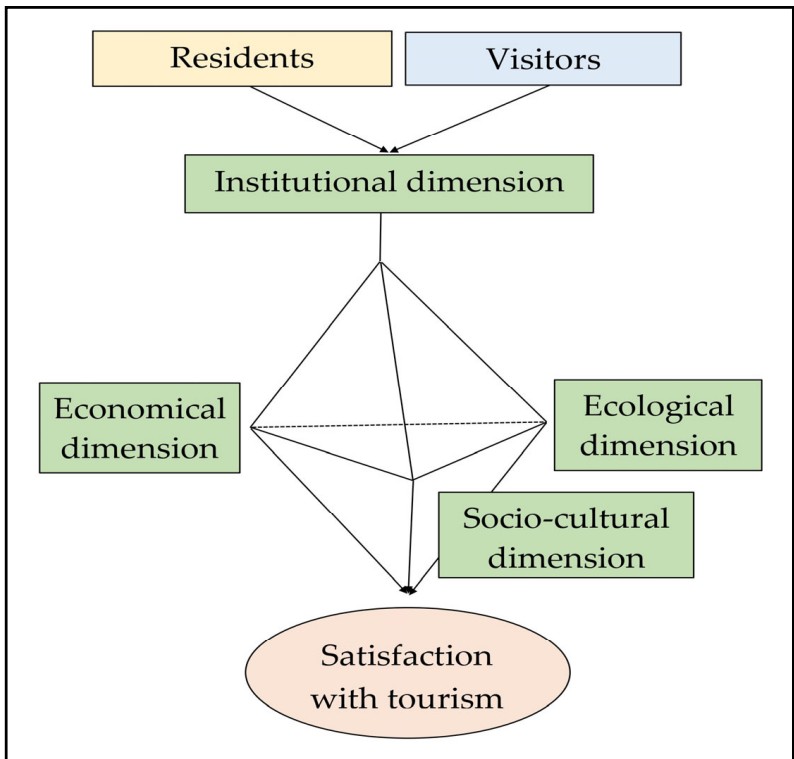

**Figure 2.** The Conceptual Model. Adapted from Cottrell et al. [54].

The main objective of this research was to collect and analyze data with the help of a quantitative methodology that can confirm or refute the research hypotheses:
Main hypothesis:

**H1.** *Sustainable tourism has a positive effect on the satisfaction of residents and visitors.*

Auxiliary hypotheses:

**H1.1.** *The institutional dimension of sustainability significantly affects the sustainable development of tourism.*

**H1.2.** *The ecological dimension of sustainability significantly affects the sustainable development of tourism.*

**H1.3.** *The economic dimension of sustainability significantly affects the sustainable development of tourism.*

**H1.4.** *The socio-cultural dimension of sustainability significantly affects the sustainable development of tourism.*

The authors planned to explore the state of sustainable tourism in a larger number of protected areas where certain tourist activities are carried out. The investigated areas were part of the geographical and territorial unit of the Republic of Serbia, which is called Vojvodina. This research is considered a continuation of the authors' previous research. It differs from other research in that sustainable tourism's impact on the satisfaction of residents and visitors in the ONL was measured by applying a quantitative methodology that included surveying respondents as a research technique. The specific lessons learnt from previous papers were that the study of sustainable tourism should be carried out on the largest possible sample by surveying both visitors and residents using the same questionnaire. Therefore, in this research, data were collected by surveying visitors and residents in the protected area. A written questionnaire was used in the survey. The results obtained by surveying residents and visitors were analyzed separately, after which the results were compared and tabulated. By conducting a comparative analysis of the obtained results, using the statistical SPSS v.21 software (IBM, Armonk, NY, USA), more reliable results pertaining to the satisfaction of the examinees could be obtained. If the comparative analysis established that similar values were obtained from both groups of respondents, this indicated the validity of the collected answers. More reliable results can indicate more significant values related to the function of this protected area in sustainable tourism. If the analysis of the data showed significant differences in the responses of the two groups of respondents, the authors would conduct individual detailed investigations of the circumstances that contributed to such responses. In addition, the authors would examine the competencies and level of perception of the respondents, which will be the subject of future research.

A questionnaire was used for this research, and the respondents were selected randomly. Visitors to the protected area were approached by a random selection of individuals or groups during the visit. The selection of residents was carried out using the same method, whereby the authors selected an individual or a household. The online questionnaire was posted on social networks within target and thematic groups. A total of 1419 respondents (789 residents and 630 visitors) were surveyed for the basis of this research. Respondents ranked their answers using a five-point Likert scale ranging from 1 (strongly disagree) to 5 (strongly agree) [9,63]. The questionnaire was designed so that it contained a total of 17 statements related to the state of sustainable tourism in the ONL, in addition to the socio-demographic characteristics of the respondents. These statements were grouped into four dimensions of sustainability. Additionally, the questionnaire contained four assertions regarding satisfaction with sustainable tourism development [54,62]. The methodology in this research used Cronbach's Alpha coefficient to test the reliability of the variables and obtained values, as well as to measure the degree of respondents' satisfaction with the four dimensions of sustainability [52,53,58]. In order to examine the degree of satisfaction of residents and visitors with the dimensions of sustainability, regression analysis was used [53]. After the values were obtained, an analysis was performed using comparison methods for both groups, which can be seen in Table 1.

**Table 1.** Respondents' understanding of dimensions of sustainable tourism (n = 1419).

| Items | Residents (n = 789) | | Visitors (n = 630) | |
|---|---|---|---|---|
| **Dimensions of Sustainable Tourism** | **α** | **Mean** | **α** | **Mean** |
| Institutional Dimension | 0.603 | 3.09 | 0.600 | 3.11 |
| Visitors are guided through the protected area by trained guides and representatives of the local community | | 3.04 | | 3.12 |
| Visitors in the protected area can see the local products (wineries, ethno-houses, handicrafts, local enterprises, etc.) | | 3.02 | | 3.09 |
| In the protected area, the manager's instructions on nature protection and visitors' activities are followed | | 3.14 | | 3.12 |
| Visitors are provided with information that reflects the history of the reserve, population, and settlements | | 3.17 | | 3.13 |
| Ecological Dimension | 0.756 | 3.46 | 0.789 | 3.84 |
| There is a joint role of visitors and residents in protecting the area | | 3.56 | | 3.80 |
| There are facilities, services, and activities available to visitors and the local community in the protected area | | 3.67 | | 4.12 |
| There are tourist facilities that do not impact the environment | | 3.14 | | 3.62 |
| Economic Dimension | 0.653 | 3.45 | 0.662 | 3.46 |
| Tourism in the protected area benefits the local community | | 3.00 | | 3.06 |
| Tourism in the protected area supports the local economy | | 3.14 | | 3.23 |
| Tourism in the protected area contributes to the employment of the local population | | 2.97 | | 2.88 |
| Local products are available to visitors | | 4.01 | | 4.00 |
| Visitors support the prices of domestic products | | 4.15 | | 4.13 |
| Socio-cultural Dimension | 0.715 | 4.01 | 0.672 | 3.99 |
| Visitors are interested in home-made products and crafts | | 4.15 | | 4.09 |
| Visitors are in contact with residents | | 3.56 | | 3.69 |
| Visitors are interested in local traditions and customs | | 4.17 | | 3.94 |
| Visitors visit local cultural attractions and events | | 4.52 | | 4.09 |
| Visitors are interested in historical sites | | 3.69 | | 4.12 |

Items measured on a 5-point Likert agreement scale; α—Cronbach's Alpha Reliability.

Written and online questionnaires were used to survey the respondents. Written questionnaires were completed in person. A total of 65% of respondents were surveyed in person. Online questionnaires were distributed with the help of social networks.

The settlements where the residents were surveyed (n = 789) were Vršac, Jablanka, Mesić, Veliko Središte, Gudurica, and Markovac. As for the territorial coverage of these settlements, they make up over 90% of the total number of settlements in the territory of the ONL. Domestic tourists make up 65% of the total number of visitors (n = 630). Countries from which foreign tourists come are Romania (19%), Bulgaria (17.5%), Croatia (14.5%), Montenegro (13.1%), Hungary (10%), Greece (5%), the Republic of North Macedonia (4.8%), Slovenia (4%), Austria (3.8%), Switzerland (3.5%), Germany (3.4%), and others (1.4%).

This research was conducted from March 2022 to March 2023. All completed questionnaires were valid for analysis. The survey was anonymous. By filling out the questionnaire, the respondents gave their consent for the obtained results to be used for scientific research purposes.

## 5. Results

The obtained survey results indicate that the majority of respondents are female (56%). Those who responded to the survey are, on average, 35 years old (from 18 to 78 years old). The majority of respondents have completed secondary education (54%), a total of 24.5% have completed primary education, 19.5% have attained higher or college education, and 2% hold a master's degree or a doctorate.

As part of the statistical data processing, the reliability of the variables was assessed in order to determine the dimensions of sustainability and residents' satisfaction with the development of tourism in the ONL [58]. Calculation of the index as a mean variable was carried out, which included each dimension (independent variables) [53,54]. Table 1 shows the average values obtained for each of the four dimensions of sustainability for both groups of respondents.

The total mean value of satisfaction with the development of sustainable tourism for both groups of respondents is 3.82 and 3.94 (Table 2).

**Table 2.** Scale items for the satisfaction index.

| Index | Residents (n = 789) | | Visitors (n = 630) | |
|---|---|---|---|---|
| | $\alpha$ | Mean | $\alpha$ | Mean |
| | 0.617 | 3.82 | 0.687 | 3.94 |
| Tourism in this protected area produces various benefits for me | | 3.54 | | 3.67 |
| It is important to me that there is sustainable tourism in this protected area | | 4.27 | | 3.92 |
| Tourism has contributed to the increased attractiveness of this protected area | | 4.31 | | 4.14 |
| I am satisfied with tourism in this area | | 3.18 | | 4.03 |

Using regression analysis, studies have been carried out to determine to what extent sustainability dimensions contribute to the satisfaction of residents and visitors in tourism development [61,64]. Additionally, regression analysis can be used to determine the validity of the investigated dimensions of sustainability [65,66]. Our assumption was supported by all four dimensional scores for assessing satisfaction with tourism, of which 34% (local population) and 37% (tourists) of the variance was explained ($R_1^2 = 0.341$; $R_2^2 = 0.372$) (Table 3).

**Table 3.** Regression analysis of satisfaction (n = 1419).

| Satisfaction with Tourism Items | Residents | | Visitors | |
|---|---|---|---|---|
| | $\beta$ [1] | *p*-Value | $\beta$ [1] | *p*-Value |
| Institutional dimension | 0.184 | 0.011 | 0.211 | 0.031 |
| Ecological dimension | 0.277 | 0.027 | 0.251 | 0.054 |
| Economic dimension | 0.156 | 0.006 | 0.201 | 0.037 |
| Socio-cultural dimension | 0.254 | 0.034 | 0.233 | 0.027 |

[1] Standardised $\beta$ value used, $R_1^2 = 0.341$; $R_2^2 = 0.372$.

## 6. Discussion

By analyzing the obtained results, it can be noticed that the obtained values are relatively identical for both groups of respondents. This indicates the validity and reliability of the obtained research results. The Cronbach's Alpha scores are 0.60 (Table 1) for the institutional dimension (four items), at very low levels of 0.76 and 0.79 for the ecological dimension (three items), 0.65 and 0.66 for the economic dimension (five items), and 0.71 and 0.67 for the socio-cultural dimension (five items). The institutional dimension of sustainability has the lowest values (3.09 and 3.11). Cotrel et al. [65] point out that an "$\alpha$" of 0.60 can be accepted as reliable in research in which there are six or fewer investigated items, although this value is at a significantly low level. If this fact is taken into account, all the tested variables can be considered reliable, i.e., all the respondents' answers can be taken into consideration. Residents and visitors gave the lowest rating to the claim that visitors can learn about the production of local products (wineries, ethno-houses, handicrafts, local enterprises, etc.). In addition, both groups of respondents gave the lowest rating to the claim that licensed tourist guides and representatives of local communities guide visitors through the protected area (3.04 and 3.12). This means that, when planning the development of

tourism in the ONL, significant participation of local community representatives in all tourist activities must be ensured. In other words, the local tourism community must be more aware of its importance and its role in the development and implementation of tourism in its territory. This leads to the creation of special tourist products and tourism development plans. In order to realize this, the local community must plan the education of the local population, as well as visitors. Among other things, this includes informing visitors about local products and how they are produced, as well as encouraging their engagement in the protection of the area. In this way, the character of institutions in the sustainable development of tourism in the ONL is strengthened, and the population has the opportunity to be directly involved in tourism development plans and the promotion of protected areas in their territory.

The socio-cultural dimension (4.01 and 3.99) and the ecological dimension of sustainability (3.46 and 3.84) have the highest average values. Thus, the obtained values indicate that these two dimensions have the greatest impact on the respondents, i.e., these dimensions of sustainability contribute to sustainable tourism development to the greatest extent. The importance of these dimensions of sustainability indicates that specific nature-based forms of tourism should be developed in the protected area of the ONL, such as ecotourism, scientific and research tourism, bird watching, nature photography, sports/recreation, excursions, and other forms of tourism. In addition, the sociocultural aspect of sustainable tourism needs to be realized through the development of events, culture, wine, gastronomy, and other forms of tourism based on the rich ethno-social motives of the population of this area. This area (the ONL) can be included nationally and internationally on maps of national and regional cultural events, and the space allows creativity and relaxation to be valued. Thus, for local producers, it is an opportunity worth capitalizing on, as it can determine sustainability and contribute to the comprehensive overview resources. Contemporary tourism needs the participation of residents, who will be able to see all the benefits brought by the development of sustainable tourism [67]. When planning tourism development, the sociocultural dimension of sustainability indicates that it is necessary to strengthen the interaction between visitors and residents. It has been observed that the creation of special forms of tourism increases chances for the positive development of tourism, while sustainable development can give the area a better position on the tourist market [68]. Therefore, in addition to the consistent application of principles aimed at the sustainable development of tourism, it is necessary to create stronger connections between the local environment and tourists, and in this way, fully apply the Prism of Sustainability model [69,70]. This can be achieved by establishing event-based and cultural tourism. Tourism products directly depend on the role of the local community in sustainable tourism. Through mutual interaction, tourists increase their understanding of local people [71]. For better development of tourism in the ONL, it is necessary to include not only residents, but also visitors. In addition, visitor centers are necessary to provide tourists with all the necessary information. The local population would work there, thus improving not only the quality of visitors' stays, but also the local population's economic status. The education of personnel for the implementation of specific tourism forms requires training and establishing a guide service in the ONL. In addition to the above-mentioned requirements, it is necessary to develop adequate facilities for the reception and accommodation of tourists in the ONL, such as ethnic households and ethnic settlements. Facilities and infrastructure must possess ecological certificates [72] and must be developed in accordance with the environmental integrity of this protected mountain area [73].

If we observe the results concerning the satisfaction of residents and visitors, we can notice relatively identical obtained values (Table 2). The average values for both groups of subjects are 3.82 and 3.94 when it comes to the satisfaction of sustainable tourism development. These values indicate the relative reliability of the obtained results. Both groups of respondents point out that it is important to develop tourism in the protected area of the ONL. Additionally, the respondents point out that the development of tourism provides various benefits. The Cronbach's Alpha scores are 0.62 and 0.69 for the satisfaction

index for both groups of respondents (Table 2). The inclusion of residents and state authorities in tourism planning and development results in benefits for residents through various activities. It is of crucial importance to develop forms of tourism in protected areas that bring visitors and residents together in mutual interaction [73]. This contributes to visitors becoming more actively acquainted with local culture and tradition [74], and gives residents opportunities to educate visitors about the importance of the protected area. Additionally, an awareness of the need to protect the space and preserve its natural characteristics is developed through mutual interaction and the exchange of experiences among all interested parties [75–78].

By analyzing the obtained average values of the dimensions of sustainability and the obtained results (Table 1), the following can be concluded: Hypothesis H1.1 (*The institutional dimension of sustainability* significantly affects the sustainable development of tourism) is partially confirmed. This indicates that it is necessary to significantly strengthen the role of institutions in the planning and growth of tourism. Hypothesis H1.2 (*The ecological dimension of sustainability significantly affects the sustainable development of tourism*) is confirmed. This important information can indicate the importance of ecological principles in tourism development, in which the primary goal is the preservation of nature. Hypothesis H1.3 (*The economic dimension of sustainability significantly affects the sustainable development of tourism*) is partially confirmed. Finally, Hypothesis H1.4 (*The socio-cultural dimension of sustainability significantly affects the sustainable development of tourism*) is fully confirmed.

Upon conducting regression analysis (Table 3), we obtained results on the significance of the levels of satisfaction of the respondents, and found that they have remarkable levels in relation to the four sustainable dimensions ($0.011 > p > 0.54$). The obtained values are relatively identical for both groups of respondents. Upon analyzing the results, the conclusions was drawn that the dimensions of sustainability significantly contribute to the overall sustainable development of tourism because they individually contribute to the satisfaction of respondents. With this, Hypothesis $H_1$ (*Sustainable tourism has a positive effect on the satisfaction of residents and visitors*) is fully confirmed.

This indicates that the protected area of the ONL has an important role in sustainable tourism development. This information could be significant when planning tourism development and creating tourist destinations [79]. This implies that managers of protected areas must consider the importance of preserving ecological principles when defining tourist activities. Plant and animal species represent a special attraction in protected areas. However, they are often fully exposed to emerging negative anthropogenic impacts. The present degradation and exploitation of these areas represent obstacles to the creation of desirable tourist destinations. In writing tourism development strategies and studies, the socio-cultural dimension of sustainability must also be included. This presupposes active perception of the local population at all stages of the development of tourism. This implies that the promotion of ethno-social values and the interaction of residents and visitors are extremely important. These can be realized through various specific forms of tourism, among which events and cultural tourism occupy an important place. The protection of nature, the possibility of consuming local products, bringing residents and visitors closer together, and the development of infrastructure without harmful effects on the environment are just some of the factors that can contribute to significant economic benefits. As tourism in the 21st century is characterized by a 'turn towards nature', sustainable tourism in protected areas is an important form of tourist movement. Proving the need to strengthen this form of tourism represents a practical application and the expected scientific contribution of this research.

## 7. Conclusions

(1)  The aforementioned research is based on the fundamental principles of sustainable development, the application and successful implementation of which increases the ONL's opportunity to succeed in developing sustainable tourism and to appear on the tourist market as a sustainable tourism destination. The need for constant education

and training of visitors, tourism staff, and the local population, and the allocation of funds for protecting and preserving the area (both its natural and anthropogenic components), should be added to these principles. The results of this research indicate that the protected ONL area has various possibilities for the development of special forms of tourism. The respondents recognized ecological and socio-cultural sustainability as the most important dimensions of sustainability. These data can be used when planning and developing different forms of tourism. Herein, above all, we mean forms of tourism based on natural resources. For the preservation of the ONL, the most significant form of tourism would be ecotourism with ecological and sustainable components. The basis of the study of eco-tourism is the preservation of nature and the protection of environments that provide a special atmosphere for tourists. Additionally, these environments are tourist products, the value of which is known to tourists who are aware of their uniqueness and who will contribute to their overall protection with their actions and the experiences they gain [79]. Certainly, the above-mentioned forms of tourism can only be developed in preserved nature in cooperation with the local community. This is why destinations whose local populations have an active role are of particular importance for sustainable tourism. In such destinations, properly developed tourism creates benefits, while the effects of development mutually interact in different ways. The ONL could be an important destination for sustainable tourism because it has specific natural and social elements that can influence the development of different forms of tourism. In addition to relief, geographical position, and hydrographic potential, the diversity and wealth of autochthonous flora and fauna are important for tourism development. These factors enable the development of education, recreation, excursion, and ecotourism as primary tourism forms. The ONL in Serbia is inhabited by extremely rare bird species, which makes it possible to organize photo safaris or ornithological tourist tours (educational forms of tourism). The population living in the ONL has rich traditions, culture, and cultural and historical heritage, as well as handicrafts, music, national dances, gastronomy, vineyards, orchards, and many other resources. It represents a rich basis for the development of different forms of tourism, such as sustainable, cultural, and event tourism. The two above-mentioned specific forms of tourism have the characteristic of incorporating social motives into their tourist offerings. Together with natural motifs, a high-quality tourist destination can be created, whose main priority would be the protection of the environment and its species.

(2) When it comes to specific tourist products, it is necessary to keep in mind that there are a wide range of services and products intended for different market segments. Tourism development can benefit all users of the protected area. Segmentation, and then, the creation of a specific product for the selected segment or segments is the basis of the market performance of specific tourism forms. An example is the organization of planned groups based on people's interests. These are small groups who are interested in scientific education and who have specific needs that can be met at the NLO. Those groups can be financed by special sources if they are concerned with research or education. In this way, they directly help to establish the financing of national parks and sustainable development, especially in developing countries [39].

(3) Marketing activities are extremely important for the promotion and development of protected areas. Accordingly, it is important that managers of protected natural assets gain experience of all marketing tools. This is why they must know that the successful implementation of ecological components, the protection of the environment, and giving priority to products that are organized in accordance with ecological standards form the basis for creating the right image [80]. For a long time, sustainable tourism has been a very important use of space, considering the benefits that result from the overall development of tourism [81].

## 8. Limitations and Future Research

When researching certain phenomena and processes, we always start from the objective of the research and strive to obtain valid results. On this occasion, what challenged the authors were the obstacles they encountered, as well as the unexpected results they obtained. The authors of this paper see all of this as beneficial to future research, as new opportunities and new chapters are opened for further work and deeper research. First of all, the obstacles that were encountered in previous research were related to restrictions created due to COVID-19, economic problems, and the risks of conflict that have arisen. These limitations were largely overcome by using the Internet and social networks, through which surveys were conducted. The results obtained from research in this and other protected areas create new ideas and open up new opportunities for in-depth research into sustainable development and the development of special forms of tourism in protected areas. Given that the Vršac Mountains and the ONL are located near the border of Serbia and Romania, future research on this protected area in Romania and on the possibilities for the cross-border cooperation of these protected areas is planned. Special emphasis will be placed on research into visitors and the local population, with the aim of the joint promotion and presentation of these supports on the national, regional, and international markets. Measuring satisfaction and finding relevant satisfaction factors that are applicable to all stakeholders involved in sustainable tourism, in any specific area, is a clear direction for future research. These satisfaction factors can be framed into a more general theory of social capital. The ideas created through this research will be used in further papers and the creation of a new concept of sustainability through tourism.

**Author Contributions:** Conceptualization, I.T., F.N., V.R., S.Š., M.M. and I.A.A.; methodology, I.T., F.N., V.R., S.Š., M.M. and I.A.A.; software I.T., F.N., M.M. and I.A.A.; validation, I.T., F.N., V.R., S.Š. and I.A.A.; formal analysis, I.T., F.N., V.R., S.Š., M.M. and I.A.A.; investigation, I.T., V.R., S.Š. and M.M.; resources, I.T., F.N., V.R., S.Š., M.M. and I.A.A.; data curation, I.T., F.N., S.Š., M.M. and I.A.A.; writing—original draft preparation, I.T., F.N., V.R., S.Š. and I.A.A.; writing—review and editing, I.T., F.N., V.R., S.Š., M.M. and I.A.A.; visualization, I.T., F.N., V.R., S.Š., M.M. and I.A.A.; supervision, I.T., F.N., V.R. and I.A.A.; project administration, I.T., F.N., V.R., S.Š., M.M. and I.A.A. funding acquisition, I.T., F.N., V.R., S.Š., M.M. and I.A.A. All authors have read and agreed to the published version of the manuscript.

**Funding:** This research received no external funding.

**Institutional Review Board Statement:** Not applicable.

**Informed Consent Statement:** Not applicable.

**Data Availability Statement:** The data that support the findings of this study are available upon reasonable request from the authors.

**Conflicts of Interest:** The authors declare no conflict of interest.

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
