# Peer review of "Sustainable Tourism in Protected Areas—The Case of the Vršac Mountains Outstanding Natural Landscape, Vojvodina Province (Northern Serbia)"

_sustainability, doi:10.3390/su15107760_

Round 1
Reviewer 1 Report
I put my comments in post-it notes in the margin.

Author Response
Respected reviewer,
Thank you for your valuable comments reviewing our work. Your comments will significantly improve the quality of the paper. Accordingly, we have fully made all the corrections you requested (lines: 53, 76, 588, 604).
Best Regards,
Authors
Reviewer 2 Report
- Usually keywords don't take (over) sequences from the title (eg. sustainable tourism and so on) - please replace them in the way to reflect the article ideas and not just be redundant
- Figure 2. The Conceptual Model. – is missing – please solve this
- The references are not in proper order / line and not all sources are correctly cited (eg. souce 41, 42, 44, 63, 65 – and so on - refer to sources that are not found in the article’s text) - please allign all of them very carefully accordingly. Besides this, references 1, 2, 10, 23, 30, 54, 56, 58, 61, 67, 71 and 80 are self-references, please? – if the answer is yes, please solve the situation, in order to avoid possible discussions related to professional ethics.
- „The introduction should briefly place the study in a broad context and highlight why it is important. It should define the purpose of the work and its significance. The current state of the research field should be carefully reviewed and key publications cited. Please highlight controversial and diverging hypotheses .... Finally, briefly mention the main aim of the work and highlight the principal conclusions. .... Please keep the introduction comprehensible to scientists outside your particular field of research” - please align with Articles template
- The section of introduction should include (even briefly at the end of the chapter): the context of the study, which are the main results presented in short, which is the originality of this paper, the main implication policy of these results and a description of the structure of the paper - the role of each section of the paper. Some of them are missing - please fill it accordingly
- The Literature Review chapter after Introduction) should include in more detail the “gap” in existing literature and the innovative aspects brought by this paper (analysis for existing literature and the novelty and originality brought by this paper should be highlighted - partially done in some chapters) - please detail the gaps in the existing literature and state more clearly / more explicitly the manner in which the article addresses these gaps.
- Would be really appreciated (actually is a request) if you wil (can) formulate (a few or at least one) Objectiv/s of the study with (minimum) two Hypotheses (e.g. Hypothessis 1, 2, 3… - introduced perhaps at the end of Introduction chapter or at the beginning of Research aria chapter) in the way to be validated by the research in Results, Discussion and Conclusions chapters (with clear reference to hypotheses 1 & 2 & (and so on) of the objective/s of the research). This would prove that there is a line of argumentation - that starts from (minimum) one objective (with minimum 2 hypothesses) and reaches a conclusion (proven with the help of the study).
- An uninformed reader can think that it is a descriptive article and that’s why, I recommend that the “concrete” proposals with “practical” applicability and if possible... “measurable” be more clearly individualized (in a separate subsection at Results’s or/and Discussion’s (sub)chapter). Actually, it would be interesting if the study would present some aspects more clearly related to the practical application of the study (examples) and its results (where could be applied, how could be applied and so on). Thus, please detail further the interpretation of the data analysis performed and its implications by reference to the scope of the research
Author Response
Respected reviewer,
We appreciate your efforts, and all the comments you wrote while reviewing our paper. Based on that, we made all the corrections in the text that you requested.
Point 1: Usually keywords don't take (over) sequences from the title (eg. sustainable tourism and so on) - please replace them in the way to reflect the article ideas and not just be redundant
Response 1: In the paper, we changed the key words and adapted them to the research (line: 36).
Point 2: Figure 2. The Conceptual Model – is missing – please solve this
Response 2: When submitting the paper and converting it to a pdf version, Figure 2 was deleted in the text. We added it again and it is now visible (line 305)
Point 3: The references are not in proper order / line and not all sources are correctly cited (eg. souce 41, 42, 44, 63, 65 – and so on - refer to sources that are not found in the article’s text) - please allign all of them very carefully accordingly. Besides this, references 1, 2, 10, 23, 30, 54, 56, 58, 61, 67, 71 and 80 are self-references, please? – if the answer is yes, please solve the situation, in order to avoid possible discussions related to professional ethics.
Response 3: We inform you that we have changed the order of references, so that they correspond to the current text. One of the reasons for this was that when converting to pdf, two paragraphs were missing, so some references were out of order. We also notify you that we have deleted most of the self references. Now there are a total of 6 of them in the paper. They are references numbered: 1,2,30,59,62 and 72.
Point 4: „The introduction should briefly place the study in a broad context and highlight why it is important. It should define the purpose of the work and its significance. The current state of the research field should be carefully reviewed and key publications cited. Please highlight controversial and diverging hypotheses .... Finally, briefly mention the main aim of the work and highlight the principal conclusions. .... Please keep the introduction comprehensible to scientists outside your particular field of research” - please align with Articles template. The section of introduction should include (even briefly at the end of the chapter): the context of the study, which are the main results presented in short, which is the originality of this paper, the main implication policy of these results and a description of the structure of the paper - the role of each section of the paper. Some of them are missing - please fill it accordingly.
Response 4: The introduction chapter has been changed significantly. We have taken into account all your suggestions. All changes in the introduction are marked with red text (lines 39-46; 94-102; 109-116; 126-152).
Point 5: The Literature Review chapter after Introduction) should include in more detail the “gap” in existing literature and the innovative aspects brought by this paper (analysis for existing literature and the novelty and originality brought by this paper should be highlighted - partially done in some chapters) - please detail the gaps in the existing literature and state more clearly / more explicitly the manner in which the article addresses these gaps.
Response 5: In the main part of the Literature review chapter, we explained what gaps existed in previous research and how we overcome the gaps in the current research. Also, we have listed and described the main shortcomings in the research and what reflects the originality of this research (lines: 154-165)
Point 6: Would be really appreciated (actually is a request) if you wil (can) formulate (a few or at least one) Objectiv/s of the study with (minimum) two Hypotheses (e.g. Hypothessis 1, 2, 3… - introduced perhaps at the end of Introduction chapter or at the beginning of Research aria chapter) in the way to be validated by the research in Results, Discussion and Conclusions chapters (with clear reference to hypotheses 1 & 2 & (and so on) of the objective/s of the research). This would prove that there is a line of argumentation - that starts from (minimum) one objective (with minimum 2 hypothesses) and reaches a conclusion (proven with the help of the study).
Response 6: We fully took into account your suggestion, so we defined the basic hypothesis and 4 auxiliary hypotheses (lines 309-321) in the research. We also tested the hypotheses and after the analysis confirmed or denied the stated hypotheses (lines 486-497; 503-504). This has significantly improved the quality of research and we are grateful to you for that.
Point 7: An uninformed reader can think that it is a descriptive article and that’s why, I recommend that the “concrete” proposals with “practical” applicability and if possible... “measurable” be more clearly individualized (in a separate subsection at Results’s or/and Discussion’s (sub)chapter). Actually, it would be interesting if the study would present some aspects more clearly related to the practical application of the study (examples) and its results (where could be applied, how could be applied and so on). Thus, please detail further the interpretation of the data analysis performed and its implications by reference to the scope of the research.
Response 7: We have taken this suggestion of yours into full consideration. We described the practical application of this research and its implications in the Discussion chapter (lines 505-524).
Best Regards,
Authors

Reviewer 3 Report
This paper has done some work. Some small issues:
1. Where Figure 2?
2. In the Introduction, the first occurrence of ONL requires explanation.
3. Redraw Figure 1.
4.The conclusions are listed with 1)... 2)... 3)...
5. Line 473 as Section 8
Author Response
We would like to thank you for your efforts and the trust you have shown in creating comments for our work. We would like to inform you that we have made all the corrections you requested in the review.
- Where Figure 2?
- In the Introduction, the first occurrence of ONL requires explanation.
- Redraw Figure 1.
4.The conclusions are listed with 1)... 2)... 3)...
- Line 473 as Section 8
Responses: When submitting the text and converting it to PDF, Figure 2 was lost. Now just put it back into the text (line 305). In the Introduction, in the first sentence, we derived the abbreviation ONL - The Outstanding Natural Landscapes Vršac Mountains (ONL) (lines 46-47). We re-entered Figure 1, it is now high resolution and not blurry. Otherwise, every photo will be slightly blurry when converted to PDF (line 274). The conclusion is divided into paragraphs 1), 2) and 3) (lines 526-581). We have added Section 8 as the last chapter (line 582).
Best regards
Authors

Reviewer 4 Report
Dear Authors,
The study is interesting and certainly makes some contribution to the field but there are quite some improvements needed before considering it for further processing. Below are some issues to be addressed:
Abstract
The abstract should be more concise and informative. As such it is wordy and do not clearly highlight the main points of the study. Some of the information like the software used should remain in the material and methods only. The introductory part would be better suited for the “Introduction” section. The rationale of the study is not well outlined. The abstract should end with more precise formulated concluding remarks that summarizes the study’s findings and contributions to the field.
Key words: should better identify the core focus of the study
Introduction
L64 “famous Vršac vineyard” – be more precise about assests of Vršac vineyard and avoid such evaluative statements.
The main aim of the study is clumsily written. It needs throughout revision because it remains unclear what you mean behind the “satisfaction” and why you introduced the “function” mentioned in L84-96 if the focus of the paper as mentioned in the title is on satisfaction. L 84“The research subject in the paper is the examination of the satisfaction of the ONL residents and visitors, which is affected by sustainable tourism, through four dimensions of sustainability” L96 “function that the ONL has in sustainable tourism”.
The section on literature review should be revised to avoid to be more concise e.g. L190-205 “The objective of the research of Huayhuaca et al., [52]….”. Summarize the main findings relevant for your research and avoid quoting so many details.
Material and methods
L256 Explain precisely how you used the model.
The methodology section is chaotic and there are several redundancies. It needs throughout revison to clearly state how the data was collected, what is behind the comparative methodology and in which way it is a continuation of the previous research as mentioned in L266. Are there any specific lessons learned from previous studies that affect the methodological approach?
L274 “Respondents were selected using a random sample method” the sampling method should be described more in detail.
The research was conducted from March 2022 to March 2023 - It is a relatively long period of data collection with many unforeseen event that could affect the satisfaction measurement. Highlight in the limitations the potential implications for the satisfaction measurement.
What I miss in the methodology is the specification of measurement instrument used and the precise information on the origin of the items in the questionnaire in relation to the approach to satisfaction measurement in the literature, including how you tackled the differences between satisfaction of residents vs visitors.
Results
The information on the concept of the measurement instrument is fragmented that makes the interpretation of the results difficult. Some statements like “There is a joint role of visitors and residents in protecting the area” lacks clarity. What is the joint role?
I am not truly convinced that the items used to measure the satisfaction of residents have much to do with their satisfaction per se.
Discussion
The discussion is very much recommendation oriented and lack clear relation to literature on satisfaction. As a consequence, conclusions should be also revised.
There are also clerical mistakes in the manuscript that need to be tackled.
Author Response
Respected reviewer,
We greatly appreciate your efforts in reviewing our paper. With that in mind, we've made all the corrections you mentioned in your review.
Point 1: The abstract should be more concise and informative. As such it is wordy and do not clearly highlight the main points of the study. Some of the information like the software used should remain in the material and methods only. The introductory part would be better suited for the “Introduction” section. The rationale of the study is not well outlined. The abstract should end with more precise formulated concluding remarks that summarizes the study’s findings and contributions to the field. Key words: should better identify the core focus of the study.
Response 1: In the paper, we changed the content of the abstract and keywords according to your suggestions (lines: 19-36).
Point 2: L64 “famous Vršac vineyard” – be more precise about assests of Vršac vineyard and avoid such evaluative statements.
Response 2: We adjusted the sentence according to your suggestion (line: 67).
Point 3: The main aim of the study is clumsily written. It needs throughout revision because it remains unclear what you mean behind the “satisfaction” and why you introduced the “function” mentioned in L84-96 if the focus of the paper as mentioned in the title is on satisfaction. L 84“The research subject in the paper is the examination of the satisfaction of the ONL residents and visitors, which is affected by sustainable tourism, through four dimensions of sustainability” L96 “function that the ONL has in sustainable tourism”.
Response 3: We inform you that we have changed the Introduction chapter and that we have defined the importance of this study, the main objective, conclusions and context of the study. Now this chapter is very clearly defined. We also deleted all the unclear parts in this chapter, in everything according to your suggestions (lines: 39-46; 94-116; 126-152).
Point 4: The section on literature review should be revised to avoid to be more concise e.g. L190-205 “The objective of the research of Huayhuaca et al., [52]….”. Summarize the main findings relevant for your research and avoid quoting so many details.
Response 4: The Literature review chapter has been significantly modified according to your suggestions. We shortened the text and left the most important elements (lines 242-249).
Point 5: L256 Explain precisely how you used the model.
The methodology section is chaotic and there are several redundancies. It needs throughout revison to clearly state how the data was collected, what is behind the comparative methodology and in which way it is a continuation of the previous research as mentioned in L266. Are there any specific lessons learned from previous studies that affect the methodological approach?
Response 5: We changed the Methodology chapter in everything according to your suggestions. We have stated how the research model was conceived and used (lines 291-301). We defined the research hypotheses, so now the research model can be clearly seen (lines 309-321). We have described the specific lessons of previous studies (lines 328-334) and the results of comparative methodology (lines 337-344).
Point 6: L274 “Respondents were selected using a random sample method” the sampling method should be described more in detail.
The research was conducted from March 2022 to March 2023 - It is a relatively long period of data collection with many unforeseen event that could affect the satisfaction measurement. Highlight in the limitations the potential implications for the satisfaction measurement. What I miss in the methodology is the specification of measurement instrument used and the precise information on the origin of the items in the questionnaire in relation to the approach to satisfaction measurement in the literature, including how you tackled the differences between satisfaction of residents vs visitors.
Response 6: We have taken your suggestion fully into account, so we have described the sampling method (lines 331-334; 346-349; 364-367). In the Introduction, we stated the limitations we had in the research (lines 149-152). We also described the origin of the questionnaire used in the research (lines 289-301).
Point 7: The discussion is very much recommendation oriented and lack clear relation to literature on satisfaction. As a consequence, conclusions should be also revised.
There are also clerical mistakes in the manuscript that need to be tackled.
Response 7: We have revised the Discussion chapter and made it clearer in relation to the research. We have divided this chapter into 3 sections. In addition, we have formed a separate Chapter 8 (lines 526, 565, 576 and 582).
Best Regards,
Authors

Round 2
Reviewer 2 Report
- You stated in the Introduction chapter that "The aim of the research was to examine the impact of sustainable tourism on the satisfaction of residents and visitors of the ONL using the survey technique, with the help of a questionnaire as a research instrument. A total of 1,419 respondents were surveyed (789 residents and 630 visitors)", and in "Data Availability Statement: Not applicable" - but "The Materials and Methods should be described with sufficient details to allow others to replicate and build on the published results. Please note that the publication of your manuscript implies that you must make all materials, data, computer code, and proto-cols associated with the publication available to readers", according to the article's template. Therefore, please specify as clearly as possible where the data / information processed in the article is available
- Also, please remove absolutely all self-citations from the text: 1., 2., 30., 59., 62.. 68., 72. and so on. I asked you before, but……Besides this, [59] is connected with [53,54] and may be considered to alter the results.
Author Response
Dear Reviewer,
We have listed the application in the text - The data that support the findings of this study are available upon reasonable request from the authors.
We removed all self-references from the text: 1, 2, 30, 59, 62, 68, and 72.
Reviewer 4 Report
L19-25 The introductory part of the abstract still wordy and does not clearly state the rationale for the study.
L39 Is really a continuation a main reason for the significance of the study? It sounds quite odd.
L94 “The main goal of this research is to examine the function of sustainable tourism within the protected area with the help of quantitative method”. The title of the paper should be modified to better reflect the main aim of the study.
Author Response
Dear reviewer,
- We arranged the text in introductory part of the abstract and added new sentences.
- The Introduction chapter has a new first sentence, it is now clearer.
- The main title has been changed according to your suggestion.
Thank you.
Round 3
Reviewer 2 Report
Dear Respectable Authors
I had the request to remove all self-citations from the text: 1., 2., 30., 59., 62.. 68., 72. and so on. I checked and noticed that you removed all self-citations from the text - without interfering with the text.
The elimination of citations also presupposed the "reformulation" of the paragraph in the article - corresponding to the quoted source. Otherwise, either the initially cited source "was not" correlated with the paragraph in the article, or the source cited later "is not" correlated with the paragraph in the article
I will give you some examples of paragraphs for which you have modified the source, but you have not changed anything in the text (are identical or cery similar):
“A special advantage is the proximity of Romania and its cities 51 as emitting areas. In addition, the diversity of flora and fauna, the rich tradition and 52 ethnic music of the population living next to the protected area, as well as the existence of 53 a significant cultural and historical heritage represent a significant factor in the tourism 54 development of the ONL protected area [1].
“The special feature of this protected area is that it is located on the Vršac 72 Mountains, exactly where Vršac vineyard is also situated (cover around 1.700 ha). Grape 73 varieties are produced here, used for the production of the quality white and red wines 74 which this area is known (Župljanka, Riesling italico, Gutedel weisser - Chasselas, and 75 Kreaca). Thus, it is possible to develop wine tourism in the ONL area, which, together 76 with other natural and social factors, can form a tourism offer made up of 77 complementary motives [2].”
“The development of specific forms of tourist activities within protected areas 201 represents a significant motive for movement, which results in tourism income. A large 202 part of the income would be refinanced precisely to improve the natural values of these 203 destinations [30,31]. VERSUS “The development of specific forms of tourist activities within protected areas 204 represents a significant motive for movement, which results in tourism income [31]. A 205 large part of the income would be refinanced precisely to improve the natural values of 206 OF these destinations”
“In examining the function of the protected area of the ONL in sustainable tourism 289 development, the Prism of Sustainability Model was used, which was designed according to 290 the models of examining sustainable tourism in other protected areas [53,54,59].”
“The socio-cultural dimension of sustainability indicates that, 447 when planning the development of tourism, it is necessary to strengthen the interaction 448 between visitors and residents, this approach being the essence of creative tourism [68]. VERSUS “The socio-cultural dimension of sustainability indicates that, 439 when planning the development of tourism, it is necessary to strengthen the interaction 440 between visitors and residents. The development of special forms of tourism increases 441 the chances of tourism development and its survival, and sustainable development 442 affects a better position on the tourist market [68]…”
Having said that, I want to inform you that absolutely I do not want to adopt a negative decision regarding the revised document, but that I am very close to reject the article in the situation you continue to treat the reviewer's recommendations in the same way or to decline my competence so, that another reviewer can make the decision in my place - because the approach seems to me (I'm not sure and that's why I don't want to express myself in this sense) tends to become a problem of professional ethics.
The choice is yours.
Until then, I will ask you to revise the text again, in the hope that you will identify the right formula to be published (as I also wish)
Thank you so much all of you in advance.
Author Response
Respected reviewer,
Thank you very much for all your efforts and for the reviews, which we took into consideration with appreciation. We follow the suggestions and made the changes to the text as requested.
(Lines: 67-74; 85-92; 230-236; 355-358; 513-520).
Thank you for your attention and understanding,
Authors.
Reviewer 4 Report
No further comments
Author Response
Respected reviewer,
Thank you for your valuable comments reviewing our research. Your comments will significantly improve the quality of the paper.
Best Regards,
Authors